# Rescheduling Plan Optimization of Underground Mine Haulage Equipment Based on Random Breakdown Simulation

**Ning Li [1,2,\*], Shuzhao Feng [1,\*], Tao Lei [1,2,\*], Haiwang Ye [1], Qizhou Wang [1], Liguan Wang [3] and Mingtao Jia [3]**

[1] School of Resource and Environment Engineering, Wuhan University of Technology, Wuhan 430070, China; yehaiwang@sina.com (H.Y.); wqz@whut.edu.cn (Q.W.)

[2] Hubei Key Laboratory of Mineral Resources Processing and Environment, Wuhan 430070, China

[3] School of Resource and Safety Engineering, Central South University, Changsha 410083, China; 13808478410@163.com (L.W.); mingtao_jia@163.com (M.J.)

[\*] Correspondence: 13875910191@163.com (N.L.); fsz2224284697@163.com (S.F.); leitao539@163.com (T.L.); Tel.: +86-138-7591-0191 (N.L.)

**Abstract:** Due to production space and operating environment requirements, mine production equipment often breaks down, seriously affecting the mine's production schedule. To ensure the smooth completion of the haulage operation plan under abnormal conditions, a model of the haulage equipment rescheduling plan based on the random simulation of equipment breakdowns is established in this paper. The model aims to accomplish both the maximum completion rate of the original mining plan and the minimum fluctuation of the ore grade during the rescheduling period. This model is optimized by improving the wolf colony algorithm and changing the location update formula of the individuals in the wolf colony. Then, the optimal model solution can be used to optimize the rescheduling of the haulage plan by considering equipment breakdowns. The application of the proposed method in an underground mine revealed that the completion rate of the mine's daily mining plan reached 83.40% without increasing the amount of equipment, while the ore quality remained stable. Moreover, the improved optimization algorithm converged quickly and was characterized by high robustness.

**Keywords:** project scheduling; underground mine; random breakdown simulation; wolf colony algorithm; multi-objective optimization

## 1. Introduction

To reduce the waste of various resources and indirectly protect the environment, the proposal of the concept of the green mine requires a scientific mining process and efficient utilization of equipment resources. Standardized and intensive mining are the main mining modes of the future. Underground production equipment is gradually developing in a large-scale and intelligent direction. Therefore, effectively improving the mining efficiency and equipment utilization will become one of the key parameters for developing green mines and improving the economic benefits of mining enterprises. With the continuous development of information technology, big data, and artificial intelligence, intelligent mining theory and technology have come a long way, thereby providing basic support for the transformation of mining enterprises from extensive to refined. Optimal preparation of production plans and reasonable equipment scheduling are important aspects of green mines and intelligent mining. However, due to factors such as the mining environment, equipment performance, and personnel quality, abnormal conditions such as equipment breakdowns and safety accidents will inevitably occur during the production process. As a result, the original scheduling plan may not be completed, which in turn affects the overall production schedule of the mine and the economic benefits of the en-

terprise. Therefore, solving the problem of production equipment rescheduling under abnormal conditions and completing the original production plan to the greatest extent have become new research hot spots in the field of green mines and intelligent mining.

Regarding the research on the short-term plan of underground mines, some scholars conceptualize it as ore blending plan [1], and the planning preparation model is constructed with the goal of maximizing profits, considering the technical and economic requirements and spatial sequence relationship in the mining process as constraints [2]. Campeau and Gamache proposed an optimization model for short-term planning. The model considers all aspects of development and production, as well as specific restrictions on equipment and workers. Moreover, the model employs a preemptive mixed-integer program to prepare the best production planning in a short time frame [3]. Based on the short-term production plan, in order to realize the compactness of underground mine production, it is necessary to schedule production equipment scientifically and efficiently, and the scheduling optimization model is constructed with the shortest total completion time as the goal [4]. To reduce the ineffective travel time of haulage equipment, Yardimci and Karpuz proposed a shortest-path optimization method for underground mine haulage road design based on the evolutionary algorithm. This method considers kinematic constraints such as minimum turning radius and maximum slope to determine the optimal haulage path [5]. For such instances, it is necessary to prepare high-quality operation planning for haulage equipment. Åstrand et al. proposed a constraint programming method that can automatically execute short-term scheduling plans, employ large-scale neighborhood search with neighborhood definitions in specific areas, and contribute to finding high-quality schedules faster [6].

On the basis of the short-term operation plan and equipment scheduling plan, in order to conform to the actual production situation as much as possible and further improve the utilization rate of equipment, scholars from various countries consider equipment breakdowns as a precondition for the preparation of the scheduling plan, and reprepare the dispatching plan of equipment. Samatemba et al. used Rstudio software to develop an algorithm and evaluate sensitive inputs of the life cycle of mining equipment. Moreover, the algorithm was used to determine, analyze, and optimize the overall equipment efficiency [7]. After the equipment performance of the haulage equipment was evaluated, the breakdown probability of the equipment was calculated by counting the number of breakdowns and constructing a random breakdown algorithm to simulate the moment of breakdown. Zandieh and Gholami investigated the mixed flow shop scheduling problem with sequence-related setup time and random machine breakdowns. The authors employed the expected completion time as the optimization goal [8]. Al-Hinai and Elmekkawy solved the problem of flexible job shop scheduling with random breakdowns based on opportunity-constrained programming [9]. Ai et al. comprehensively considered random factors such as unit breakdowns, line breakdowns, load forecast errors, and interruptible load defaults. Moreover, the authors took the minimum system operating cost as the objective function to construct an optimization model with interruptible loads [10]. Based on the internal relationship between the scheduling scheme structure, machine breakdown probability, maintenance time, and scheduling robustness, Ba et al. proposed an alternative measurement method based on the propagation of process breakdown effects. The authors wanted to investigate the problem of measuring the robustness of job shop scheduling in a random machine breakdown environment [11].

According to the existing research of scholars from various countries, the main equipment rescheduling research work can be carried out in three steps. First, the breakdown problem is considered [12–14], and the original plan is reprogrammed [15]. Second, the existing rescheduling model is employed to construct a new rescheduling model [16], or corresponding rescheduling strategies [17–21] and frameworks [22] are proposed according to different breakdown problems. When proposing the rescheduling strategy, some researchers proposed that a method can be used to determine whether rescheduling is required [23]. Finally, based on completing the construction of the rescheduling model,

the corresponding optimization algorithm [24–27] is improved to solve the strategy model.

Existing research mostly involves the problem of assembly line production, which is very different from the actual production in mines. Based on the existing research, in order to ensure the completion of the original operation plan on time, the multi-equipment is rescheduled, and an optimization method for an underground mine haulage equipment rescheduling plan based on random breakdowns simulation is proposed, aimed at the scheduling problem in the case of the breakdown of the underground mining equipment. The main tasks of the method are:

1. Constructing a random breakdown algorithm, determining the time of breakdown occurrence and the maintenance end, and classifying and segmenting those times according to whether each breakdown time is intersected or not.
2. Constructing a rescheduling plan model during each equipment breakdown period. Here, the main goal of minimum ore grade deviation and the secondary goal of maximum completion rate of the planning during the rescheduling period are taken into consideration. Finally, the rescheduling plan model is solved via the improved wolf pack algorithm.
3. Comparing the solution results of the rescheduling plan optimization model with the traditional scheduling plan to verify its feasibility.

## 2. Model

### 2.1. Prerequisites and Problem Description

2.1.1. Prerequisites

By considering the adjustment strategy for equipment breakdowns in the actual mining process, the following assumptions are made:

1. When the haulage equipment breaks down, there is no spare equipment for dispatching, and the malfunctioning equipment can only continue to operate after the repair is completed.
2. During the daily planning period, the haulage equipment can only be operated in a single sublevel.
3. When the equipment breaks down, to avoid affecting the haulage of the line, auxiliary equipment is used to relocate it to a free area for maintenance.
4. After the equipment maintenance is completed, the start time of the equipment's continued operation is designated according to the time required to finish this operation.
5. During the daily planning period, each piece of equipment breaks down no more than once.

2.1.2. Problem Description

When the mine adopts the caving method for mining, it mainly relies on scrapers and electric locomotives for ore haulage. The task of the scraper is to transport the ore from the stope to the ore pass in a sublevel. The electric locomotive is responsible for transporting the ore from the ore pass to the main shaft.

Since the electric locomotive runs relatively slowly and travels on a fixed and smooth track, its probability of breakdown is relatively low. Compared with electric locomotives, the working environment of scrapers is more complicated. Humid air and bumpy roads increase the breakdown probability of the equipment. In addition, long-term turns and gear changes may cause fatigue and deformation of the machine parts, thereby increasing the risk of a potential breakdown.

According to the breakdown repair time of each piece of equipment, two rescheduling strategies are proposed: inserting rescheduling strategy and complete rescheduling strategy. For the former, when a certain piece of equipment breaks down, the original

scheduling plan in this period is reprepared during the maintenance period of the equipment. For the latter, when a certain piece of equipment breaks down, all subsequent uncompleted original scheduling plans are reprepared during the maintenance period of the equipment. Consequently, all subsequent uncompleted original scheduling plans are still reprepared after the equipment maintenance is completed.

When the breakdown repair time of each piece of equipment intersects on the timeline, the breakdown time is divided into multiple rescheduling periods. This is done by taking the crossing point between the breakdown time of each piece of equipment and the time when the breakdown occurs and maintenance ends. Then, the inserting rescheduling plan preparation is implemented within each rescheduling period. When the breakdown repair time of each piece of equipment does not intersect on the timeline, all subsequent periods are divided into multiple complete rescheduling periods starting from the moment of the first breakdown. This is done when the equipment is characterized by the first breakdown within the daily planning period. In such instances, the complete rescheduling plan is prepared within each rescheduling period (Figure 1).

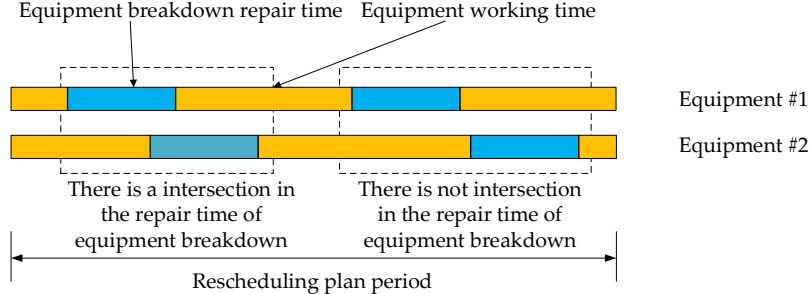

**Figure 1.** The intersection of each equipment breakdown repair time on the timeline.

The problem of rescheduling haulage equipment is described as follows. The mine point is denoted as $C_i$ and the dump point as $X_j$. The haulage equipment $M_k$ runs between the mine point and the dump point, $i = 1,2, \ldots, m, j = 1,2, \ldots, n, \; k = 1,2, \ldots, l$. Parameter $G_k$ represents the moment when the $k$-th equipment fails. $D_{kij}$ and $D_{kji}$ represent that the $k$-th equipment is located between the $i$ mine point and the $j$ dump point, respectively, representing $i \rightarrow j$ and $j \rightarrow i$. Lastly, $W_k$ denotes the maintenance time of the $k$-th equipment. The preparation of the rescheduling plan is done by rescheduling the rest of the normal operating equipment within the specific time range. To meet the original scheduling task as much as possible, this time range is defined from the breakdown of each piece of equipment to the end of the maintenance.

The rescheduling plan cycle is defined as a single day, while the time period is defined as 30 min. The probability of breakdown of the haulage equipment at each time period is statistically analyzed. The random breakdown algorithm is described as follows:

1. Divide the single day into 48 parts, $t \in [1, 48], t = 1$.
2. Choose a random number $R_{tk}$ for each period which represents the random number for the $t$-th period of the $k$-th equipment.
3. Compare this random number with the breakdown probability $P_{tk}$ of the $k$-th equipment in the $t$-th period. If $R_{tk} > P_{tk}$, then the $k$-th equipment does not break down in the $t$-th period. Otherwise, it is considered that a breakdown has occurred.
4. The specific breakdown time point is randomly obtained from the planning within the $t$-th period.
5. When a breakdown occurs in the $t$-th period, the repair end time is $W_k + 0.5t$. Then, the $t'$-th period, which is the repair end period, is determined.

6. The $t$ starts from $t'+1$, and step (2) is repeated. Finally, the occurrence times for all breakdowns and the end of maintenance for each equipment within the daily planning period are determined.

*2.2. Model Building*

A strategy of inserting a rescheduling plan is proposed to consider simultaneously repairing two or more pieces of equipment. For such instances, the simultaneous breakdown of multiple pieces of equipment will seriously affect the execution of the original planning. Therefore, it is necessary to reprepare the unfinished original planning during the equipment repair period.

If the breakdown repair time of each piece of equipment does not intersect, i.e., there is only one piece of equipment that is being repaired, the impact during the repair period is smaller. Therefore, a complete rescheduling plan strategy is proposed to reprepare subsequent uncompleted original planned tasks.

2.2.1. Sets

$\mathcal{I}$: Set of stopes that participated in the original planning.

$$\mathcal{I} = \{1, \dots, I\}, i \in \mathcal{I}$$

$\mathcal{J}$: Set of ore passes that participated in the original planning.

$$\mathcal{J} = \{1, \dots, J\}, j \in \mathcal{J}$$

$\mathcal{M}$: Equipment sets that participated in the original planning.

$$\mathcal{M} = \{1, \dots, M\}, m \in \mathcal{M}$$

$\mathcal{C}$: Set of inserting rescheduling period.

$$\mathcal{C} = \{1, \dots, C\}, c \in \mathcal{C}$$

$\mathcal{W}$: Set of complete rescheduling period.

$$\mathcal{W} = \{1, \dots, W\}, w \in \mathcal{W}$$

$\mathcal{T}$: Set of total rescheduling period that contains inserting and complete rescheduling periods.

$$\mathcal{T} = \{1, \dots, T\}, t \in \mathcal{T}$$

2.2.2. Parameters

$C_m$: Capacity of the $m$-th equipment;
$D_m$: Full bucket factor of the $m$-th equipment;
$S_i$: Ore dispersion of the $i$-th stope;
$YP_t$: Grade of the original in the $t$-th rescheduling period;
$Yore_t$: Total amount of ore mined in the $t$-th rescheduling period;
$p_i$: Grade of the $i$-th stope;
$z_{mij}$: The loaded travel time of $m$-th equipment from $i$-th stope to $j$-th ore pass;
$q_{mji}$: The unloaded travel time of $m$-th equipment from $i$-th stope to $j$-th ore pass;
$T_c$: Track of the $c$-th inserting rescheduling period;
$T_w$: Track of the $w$-th complete rescheduling period;
$v_{ci}$: The amount of ore hauled from the $i$-th stope in the original planning of the $c$-th inserting rescheduling period;
$v_{wi}$: The amount of ore that has not been hauled out of the $i$-th stope in the original daily planning during the $w$-th complete rescheduling period;
$u_{cj}$: The amount of ore to be hauled to the $j$-th ore pass in the original planning of the $c$-th inserting rescheduling period;

$u_{wj}$: The amount of ore hauled to the $j$-th ore pass that was not completed in the original daily planning during the $w$-th complete rescheduling period.

### 2.2.3. Variables

$n_{cmij}$: The number of trips of $m$-th equipment from $i$-th stope to $j$-th ore pass in the $c$-th inserting rescheduling period;

$n_{cmji}$: The number of trips of $m$-th equipment from $j$-th ore pass to $i$-th stope in the $c$-th inserting rescheduling period;

$n_{wmij}$: The number of trips of $m$-th equipment from $i$-th stope to $j$-th ore pass in the $w$-th complete rescheduling period;

$n_{wmji}$: The number of trips of $m$-th equipment from $j$-th ore pass to $i$-th stope in the $w$-th complete rescheduling period.

### 2.2.4. Rescheduling Plan Model

In the actual production process of underground mines, equipment breakdown has a significant impact on the execution of the production tasks, as well as the ore grade in the original scheduling planning. Therefore, the maximum completion rate of the ore mining planning and the smallest ore grade fluctuation in each rescheduling period are set as the desired goals. In addition, the dressing plant has high requirements for the ore grade. Therefore, during each rescheduling period, the main goal is to minimize the fluctuation of the ore grade. The maximum completion rate of the planned ore mining volume is set as the secondary goal.

According to the intersection of the breakdown repair period on the timeline, the rescheduling plan model is divided into the inserting and complete rescheduling plan models. The variables and parameters are diverse in different rescheduling plan models.

1. Inserting rescheduling plan mode

The main objective:

$$F1^1 = \left| \left( \frac{\sum_{\mathcal{M}} \sum_{\mathcal{J}} \sum_{\mathcal{J}} n_{cmij} C_m D_m S_i p_i}{\sum_{\mathcal{M}} \sum_{\mathcal{J}} \sum_{\mathcal{J}} n_{cmij} C_m D_m S_i} \right) - YP_t \right| \tag{1}$$

The secondary objective:

$$F2^1 = \left( \sum_{\mathcal{M}} \sum_{\mathcal{J}} \sum_{\mathcal{J}} n_{cmij} C_m D_m S_i \right) / Yore_t \tag{2}$$

Constraints:

$$\sum_{\mathcal{M}} \sum_{\mathcal{J}} n_{cmij} C_m D_m S_i \leq v_{ci}, \forall c, i \tag{3}$$

$$\sum_{\mathcal{M}} \sum_{\mathcal{J}} n_{cmij} C_m D_m S_i \leq u_{cj}, \forall c, j \tag{4}$$

$$\sum_{\mathcal{J}} \sum_{\mathcal{J}} \left( n_{cmij} z_{mij} + n_{cmji} q_{mji} \right) \leq T_c, \forall c, m \tag{5}$$

2. Complete rescheduling plan model

The main objective:

$$F1^2 = \left| \left( \frac{\sum_{\mathcal{M}} \sum_{\mathcal{J}} \sum_{\mathcal{J}} n_{wmij} C_m D_m S_i p_i}{\sum_{\mathcal{M}} \sum_{\mathcal{J}} \sum_{\mathcal{J}} n_{wmij} C_m D_m S_i} \right) - YP_t \right| \tag{6}$$

The secondary objective:

$$F2^2 = \left( \sum_{\mathcal{M}} \sum_{\mathcal{J}} \sum_{\mathcal{J}} n_{wmij} C_m D_m S_i \right) / Yore_t \tag{7}$$

Constraints:

$$\sum_{\mathcal{M}} \sum_{\mathcal{J}} n_{wmij} C_m D_m S_i \leq v_{wi}, \forall w, i \tag{8}$$

$$\sum_{\mathcal{M}} \sum_{\mathcal{J}} n_{\mathrm{wmij}} C_{\mathrm{m}} D_{\mathrm{m}} S_{\mathrm{i}} \leq u_{\mathrm{wj}}, \forall w, j \tag{9}$$

$$\sum_{\mathcal{J}} \sum_{\mathcal{J}} \left( n_{\mathrm{wmij}} z_{\mathrm{mij}} + n_{\mathrm{wmji}} q_{\mathrm{mji}} \right) \leq T_{\mathrm{w}}, \forall w, m \tag{10}$$

The constraint shown in Equation (3) determines that the tonnage of ore hauled out of each stope during a single inserting rescheduling period cannot be greater than the tonnage of ore hauled out during that period in the originally planned task. The constraint shown in Equation (4) determines that the tonnage of ore hauled to each ore pass during a single inserting rescheduling period cannot be greater than the original daily scheduled task. The constraint shown in Equation (5) restricts the schedule duration of each piece of equipment during a single inserting rescheduling period, i.e., it cannot be greater than the duration of this period. The constraint determined by Equation (8) indicates that the tonnage of ore hauled out of each stope in a complete rescheduling period cannot be greater than the tonnage of ore that has not been hauled out of the stope in the original plan. The constraint shown in Equation (9) determines that the tonnage of ore hauled to each ore pass in a complete rescheduling period cannot be greater than the tonnage of ore hauled to the ore pass that was not completed in the original daily plan. Finally, the constraint determined by Equation (10) restricts the schedule duration of each piece of equipment in the complete rescheduling period, i.e., it cannot be greater than the duration of this period.

## 3. Optimization Algorithm

The wolf colony algorithm based on the intelligence of the wolf colony can quickly find the optimal solution, with a large probability, with a random probability search in multiple points simultaneously, thereby simulating the predatory behavior of the wolf colony and its prey distribution method. This method abstracts three intelligent behaviors: wandering, rushing, and sieging. The wolf generation rule of "the winner takes all," as well as the wolf colony update mechanism of "only the strong survive," are employed. This algorithm is characterized by high solution accuracy, fast convergence speed, few control parameters, and strong robustness. However, its disadvantage of falling into a local optimum requires enhancing the global search ability and increasing the diversity of the population to find the optimal solution in the global scope.

The wolf colony algorithm divides the population of an artificial wolf colony into three types of wolves: head wolves, detective wolves, and fierce wolves. In the solution space, the artificial wolf with the best fitness is the head wolf. The best section of artificial wolves (apart from the head wolf) in the solution space are regarded as detection wolves. The remaining population is designated as the fierce wolves. The head wolf is responsible for the command and maintenance of the entire wolf colony. The detective wolves hunt within the possible range of their prey, make independent decisions based on the scent left by the prey, and always search in the direction with the strongest smell. The fierce wolves are responsible for sieging the prey when the detective wolves find it on the trail.

### 3.1. Algorithm Performance Testing

The performance of the wolf colony algorithm is compared with the particle swarm optimization (PSO), the genetic algorithm (GA), and the gravitational search algorithm (GSA). The method is to optimize and solve Formula (11), the range of variable is [–10, 10], and the theoretical minimum value is 0. The optimal fitness value of each generation will deviate in every algorithm due to the randomness of the initial population. The fitness value can be ignored when analyzing the performance of each algorithm. Compared with other algorithms, the wolf colony algorithm converges faster in the early stage, and after the convergence, the variation of the curve is relatively smooth (Figure 2).

$$minf = 0.26(x_1^2 + x_2^2) - 0.48x_1x_2 \tag{11}$$

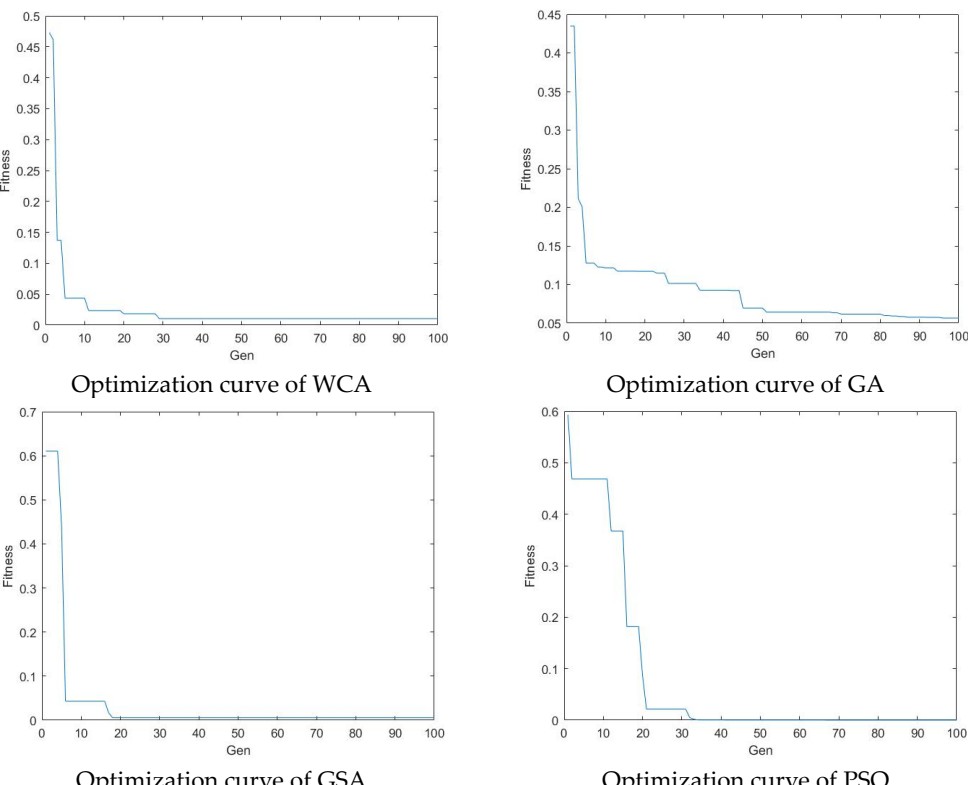

**Figure 2.** The optimization performance comparisons of various algorithms.

### 3.2. Algorithm Design and Improvement

#### 3.2.1. Population Initialization

The equipment rescheduling plan problem is a typical non-deterministic polynomial (NP) problem. When solving this type of problem, the chromosome encoding method in the genetic algorithm is preferred for describing the solution of the problem. Therefore, the chromosome encoding method is employed to initialize the wolf colony. The combination of multiple chromosomes represents a feasible solution to the problem. The chromosome length is determined according to the problem scale and can be adjusted.

Lowercase letters are used for stopes, while capital letters are used for ore passes. For example, [aAbBcC] represents the operating route of a scraper. Multiple chromosomes form a single individual in the wolf colony, thereby indicating the current location of an artificial wolf. The position of the head wolf represents the position of the prey. For equipment rescheduling planning problems, the operation route of the scraper can be coded in periods. Each chromosome represents the operation route of the scraper in a specific period. After determining the chromosome coding method, the initial population is randomly generated according to the constraint conditions.

#### 3.2.2. Rules for Determining the Head Wolf

First, the calculation method of individual fitness value is determined. The optimization model for equipment rescheduling planning is to minimize the objective function. Therefore, the reciprocal of the objective function is taken as the fitness of the individual. In addition, multiple objective functions correspond to multiple fitness values:

$$fitness1 = \max\left(\frac{1}{F1}\right) \tag{12}$$

$$fitness2 = \max(F2) \tag{13}$$

Since the objective function is not unique, and primary as well as secondary goals exist, the first fitness value of each artificial wolf in the solution space is compared against

the head wolf rules. If there are multiple artificial wolves with the optimal first fitness values, the second fitness value is compared. If multiple equal second fitness values are present, one artificial wolf is randomly selected as the head wolf among all artificial wolves with equal first and second fitness values. The head wolf does not execute three intelligent behaviors. Moreover, it directly enters the next iteration.

### 3.2.3. Wandering Behavior

The wandering behavior of detective wolves determines the global search ability of the wolf colony algorithm. With an increase in the number of detective wolves, the chance of finding the global optimal solution is also increased. All artificial wolves in the solution space are first sorted. Then, better artificial wolves with half the population size (apart from the head wolf) are considered as detective wolves. First, prey concentration $Y_i$ at the current position of the *i*-th detective wolf is calculated. If $Y_i$ is higher than the prey concentration $Y_{\text{lead}}$ perceived by the head wolf, then $Y_{\text{lead}} = Y_i$. The *i*-th detective wolf initiates the call by replacing the wolf. If $Y_i \leq Y_{\text{lead}}$, then the *i*-th detective wolf takes one step forward in *h* directions and records the perceived prey concentration after each step before returning to the original position.

The detective wolf can advance in *h* directions by randomly selecting *h* gene positions in all chromosomes of the individual detective wolf. The step length is set as two. In other words, each time detective wolves wander a single step forward, the number of genes that are exchanged between each side of the selected gene position is increased by one.

For example, the position update method of wandering forward in the *h*-th direction for two steps (Figure 3) is:

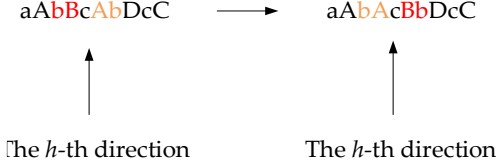

**Figure 3.** The position update method of wandering forward in the *h*-th direction for two steps.

To prevent the algorithm from falling into a local solution, when detective wolves wander in *h* directions and are still in place after reaching the predetermined number of wandering, the detective wolves choose to move to a certain position in *h* directions via the roulette method. Only *fitness*1 is selected in the roulette method.

### 3.2.4. Rushing Behavior

When detective wolves initiate a call, or the detective wolves fail to update the head wolf during the maximum number of wandering, the head wolf initiates a call. Once a call is initiated, all fierce wolves begin to rush towards the position of the head wolf. During that time, if the prey concentration $Y_i$ perceived by the *i*-th fierce wolf is greater than $Y_{\text{lead}}$, then $Y_{\text{lead}} = Y_i$. After all fierce wolves have completed rushing, the fierce wolf will transform into the head wolf and re-initiate a call. The rushing number is not calculated repeatedly. If $Y_i \leq Y_{\text{lead}}$, the *i*-th fierce wolf continues to rush. When the maximum number of rushing is reached or the distance between the fierce wolf and the head wolf is less than *d*, the sieging behavior is initiated.

The rushing method of the fierce wolves is as follows. First, the rushing step length is set as two. Then, the gene exchange position *j* of each chromosome of the *i*-th fierce wolf is randomly selected, and a single gene on each side of the gene position *j* is exchanged with some position of the head wolf. The number of exchanged genes increases with the number of rushing.

The location update method for the second rushing of the *i*-th fierce wolf (Figure 4) can be expressed as:

The *i*-th fierce wolf                 aAbBcAbDcC ⟶ aAaAcCbDcC

The head wolf                              cAaAcCbDcC

**Figure 4.** The location update method for the second rushing of the *i*-th fierce wolf.

The distance $d_i$ between the fierce wolves and the head wolf can be calculated as:

$$d_i = |fitness1_i - fitness1_{lead}| \tag{14}$$

### 3.2.5. Sieging Behavior

When the fierce wolves get closer to the prey, they should unite with the detective wolf to conduct a close sieging on the prey and capture it. The position of the head wolf is regarded as the moving position of the prey.

The sieging method of artificial wolves is as follows. The sieging step is set to two. When each artificial wolf in the wolf colony participates in the sieging behavior, a gene position is first randomly determined. Then, the position of the artificial wolf is updated according to the location update method of the rushing behavior.

### 3.2.6. "The Winner Takes All" Wolf Generation Rules

The prey is distributed according to the principle of "from strong to weak" resulting in weak wolves being eliminated. To maintain the population diversity of the wolf colony, 40% of the artificial wolves with poor fitness are eliminated. Lastly, new artificial wolves are regenerated.

### *3.3. Algorithm Flow*

The specific operation flow of the algorithm is shown in Figure 5.

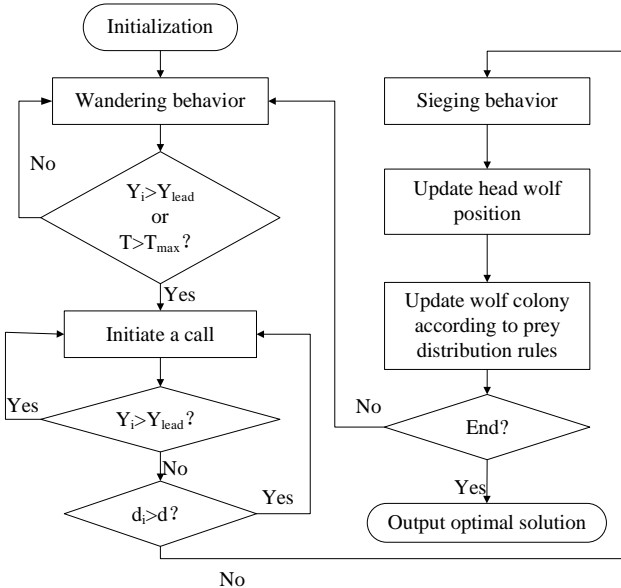

**Figure 5.** The improved wolf colony algorithm flow chart.

## 4. Computational Study

### *4.1. Basic Data of the Original Scheduling Plan*

The grades of the stopes currently being mined within a mine using the caving method are shown in Table 1. The loaded and empty running hours of the scraper between the stope and ore pass are shown in Tables 2 and 3. Part of the original scheduling plan

tasks of each scraper on a certain day are shown in Table 4, in which lowercase letters indicate the number of stopes, and capital letters indicate the number of ore passes. For example, "cAd" means that the scraper hauls ore from "c" stope to "A" ore pass, and then drives to "d" stope for the next loading. During a certain production period, there are 7 stopes, 2 ore passes, and 3 scrapers in this sublevel. The equipment capacity of the scraper is 5 t, the full bucket coefficient is 0.95, and the ore block dispersion is 0.83. A simple demonstration of the running route of the #1 scraper at a certain time is shown in Figure 6.

**Table 1.** The grade table of each stope.

| Number of Stopes | Grade/% |
|---|---|
| a | 24.03 |
| b | 61.74 |
| c | 44.26 |
| d | 36.46 |
| e | 49.15 |
| f | 49.28 |
| g | 56.74 |

Note: The data is theoretical.

**Table 2.** The loaded travel time of the scraper between the stope and ore pass (Units: seconds).

| Ore Pass | Stope/(Number) | | | | | | |
|---|---|---|---|---|---|---|---|
| | a | b | c | d | e | f | g |
| | (1,1,4,3) | (1,1,8,3) | (1,1,11,3) | (1,1,1,2) | (1,1,6,2) | (1,1,9,1) | (1,1,12,1) |
| A | 75.5 | 149.3 | 179.2 | 97.8 | 100.5 | 145.6 | 210.3 |
| B | 154.2 | 85.6 | 96.2 | 205.2 | 137.3 | 79.7 | 100.5 |

Note: The data is theoretical.

**Table 3.** The empty travel time of the scraper between the stope and ore pass (Units: seconds).

| Ore Pass | Stope/(Number) | | | | | | |
|---|---|---|---|---|---|---|---|
| | a | b | c | d | e | f | g |
| | (1,1,4,3) | (1,1,8,3) | (1,1,11,3) | (1,1,1,2) | (1,1,6,2) | (1,1,9,1) | (1,1,12,1) |
| A | 52.49 | 105.7 | 147.5 | 64.5 | 67.8 | 104.3 | 172.6 |
| B | 110.7 | 57.5 | 61.3 | 144.2 | 84.6 | 54.0 | 68.2 |

Note: The data is theoretical.

**Table 4.** Part of the operation route of the scraper.

| Number | Operation Route |
|---|---|
| 1 | cAdBaBdAeAeBaAfBaBeAfBfAcBaBdAfBgAfAdBgBbBcAb |
| 2 | BfAbBfBaBfBaAcAgBdAgAaBaAeAdBfAcAcAfAgBgAfAfBdBaAaAbA |
| 3 | AeAcAfAcAgAfAdBcAdBeBaBgAcBdAgAgAfBdBbAfAgBfAbAeB |

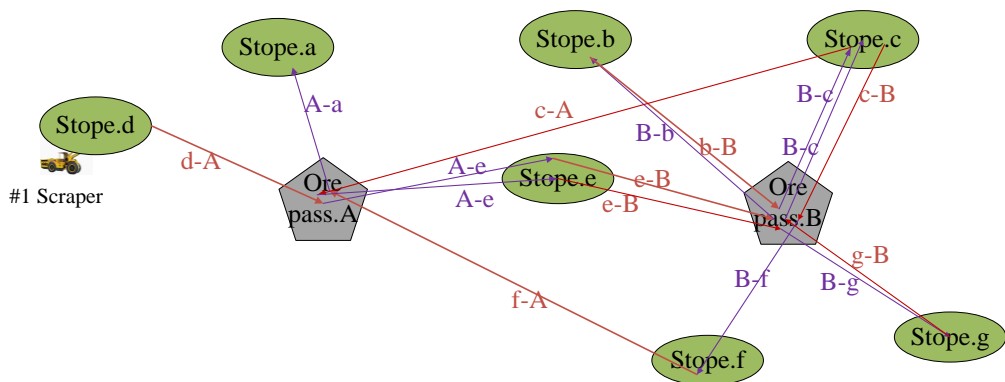

**Figure 6.** The operation route of the #1 scraper during a certain period.

The tonnage of daily ore mined in each stope, the grade of daily ore removal, and the tonnage of ore hauled to the ore passes can be calculated according to each scraper's plans (Tables 5 and 6).

**Table 5.** The tonnage of ore mined in each stope and the grade of daily ore removal.

| Number of the Stope | The Tonnage of Ore Mined/t | Grade of Daily Ore Removal/% |
|:---:|:---:|:---:|
| a | 512.53 | |
| b | 461.27 | |
| c | 595.32 | |
| d | 646.57 | 45.94 |
| e | 544.07 | |
| f | 642.63 | |
| g | 666.28 | |

Note: The data is theoretical.

**Table 6.** The tonnage of ore hauled to each ore pass.

| Number of Ore Pass | The Tonnage of Ore Hauled to Ore Pass/t |
|:---:|:---:|
| A | 2310.31 |
| B | 1758.35 |

*4.2. Equipment Breakdowns Simulation Analysis*

During the actual production process, the equipment cannot operate continuously, and the occurrence of breakdowns will lead to the inaccurate completion of the originally planned tasks. The time node of the equipment breakdown cannot be predicted. However, equipment breakdown must be considered when preparing the plan. Therefore, it is necessary to simulate the breakdown of the scraper during the operation according to the breakdown rate of the equipment. Then, based on the simulation results, the equipment rescheduling plan is prepared.

It is assumed that the change curve of the haulage equipment breakdown probability over time is $P_{tk} = 1 - 1.05^{-t}, t \geq 1$, where $t$ is an integer. Furthermore, it is assumed that the change curve can accurately simulate the occurrence of breakdowns, while the breakdown repair time is taken as four hours. The breakdown time of each piece of equipment, as well as the end time of breakdown maintenance, can be obtained through the random breakdown algorithm (Table 7).

**Table 7.** The simulated time of breakdown and maintenance end in each equipment.

| Number | Time of Breakdown | Time of Breakdown Maintenance End |
|---|---|---|
| 1 | 3:12 | 7:12 |
| 2 | 7:7 | 11:7 |
| 3 | 4:51 | 8:51 |

The analysis of each period is shown in Figure 7, where periods 1, 2, 3, 4, and 5 are denoted as inserting rescheduling periods. When the rescheduling plan is prepared, each period is carried out in order. The next period can be prepared only after the rescheduling plan of the previous period is optimized.

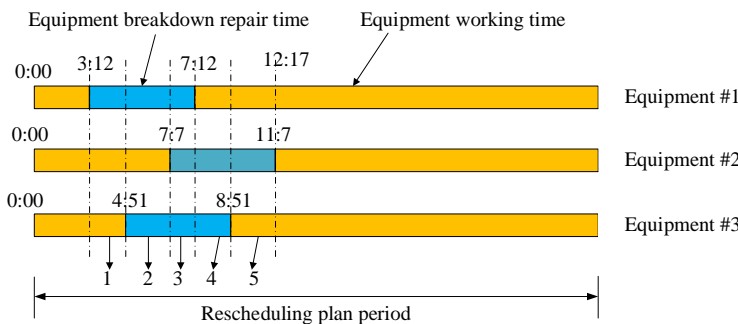

**Figure 7.** The analysis and demonstration of each period.

### 4.3. Optimization Results

The rescheduling plan of each rescheduled period is modeled according to the sequence of each rescheduled period on the timeline. The procedure is as follows: Calculate the constraint variable value of the model in the first rescheduling period, then use the improved wolf pack algorithm to solve the model. Use the solution result to update the constraint variable of the model in the next rescheduling period, and continue to solve the model. All rescheduling period models are solved according to the above method, and then all solution results are integrated into a whole rescheduling plan optimization scheme.

The number of original plan tasks in each period is taken as the constraint, while the ore grade deviation and the ore output in each period are taken as the target. Then, to compile the improved wolf colony algorithm code, the optimization plan of the rescheduling plan within each period is calculated via MATLAB software. In the wolf colony algorithm, the initial population size is 40 and the evolutionary algebra is 200. The wandering step length, the rushing step length, and the sieging step length are all set as two. The distance threshold d between the fierce wolf and the head wolf is based on the first fitness of the head wolf of each generation. This value can be set up to ten times lower than the first fitness value of the head wolf. When solving the planning model in each rescheduling period, the population must be reinitialized and parameters reset. The final solution results for each period are shown in Tables 8–11. During the third rescheduling period, all of the equipment is malfunctioning. Therefore, there is no need to prepare a rescheduling plan. To obtain a whole rescheduling plan for each piece of equipment, the solution results of each rescheduling period are inserted into the original plan. Then, the total tonnage of mined ore, the grade of daily ore removal, and the tonnage of ore hauled to each ore pass can be obtained (Tables 12 and 13).

**Table 8.** The first rescheduling period plan optimization scheme.

| Number | Operation Route | Fitness1 |
|---|---|---|
| 2 | AgBgAdBeAcBdBdAfAfBgBfAcBbAfBfAg BgBfBbAeAfAaBdBeAfBeBeAb | 1.95 × 10⁻³ |
| 3 | BdAcAcAgBeAgBcAgAdAfAdBbBgAbAeBcA bBeBgBgBfBeAeBbAa | |

**Table 9.** The second rescheduling period plan optimization scheme.

| Number | Operation Route | Fitness1 |
|---|---|---|
| 2 | BcBbBaBbBcAbAfBcAeBfAaBdBgBcAeBfAfB cAfAfAcBfAbBcAfBfBbBbBbBeAbBgBgBcAcBgBcAfBcAc | 1.075 × 10⁻⁴ |

**Table 10.** The fourth rescheduling period plan optimization scheme.

| Number | Operation Route | Fitness1 |
|---|---|---|
| 1 | cBgBbAbAcAbAcBaAcAfAeAgBbAcBcA bBcBfBcAgBdAdAfBgAdBgAgBfBf | 4.45 × 10⁻⁴ |

**Table 11.** The fifth rescheduling period plan optimization scheme.

| Number | Operation Route | Fitness1 |
|---|---|---|
| 1 | AfBfBfAeBcBeBeBfAgBbBgAgAfAgBfBaAbBaA gAdBgBcBgAaBfAfAfAcBgAcBcBbBcAfAcAfAdA | 1.51 × 10⁻² |
| 3 | BcAcAfAfBfBcAbAgBgBgBbAbAeBeBbBgBdBdBd BfAeAeBfBcBfAeAbAcAcAdAfBcBaBaBdB | |

**Table 12.** The tonnage of ore mined in each stope and the grade of daily ore removal in the rescheduling plan.

| Number of the Stope | The Tonnage of Ore Mined/t | Grade of Daily Ore Removal/% |
|---|---|---|
| a | 370.59 | |
| b | 417.91 | |
| c | 536.18 | |
| d | 488.87 | 46.68 |
| e | 441.56 | |
| f | 591.37 | |
| g | 571.66 | |

**Table 13.** The tonnage of ore hauled to each ore pass in the rescheduling plan.

| Number of Ore Pass | The Tonnage of Ore Hauled to Ore Pass/t |
|---|---|
| A | 1864.80 |
| B | 1553.34 |

*4.4. Result Analysis*

The rescheduling plan calculated by the optimization algorithm is compared with the traditional breakdown scheduling plan (Figure 8). The total ore tonnage of each stope in the rescheduling plan with WCA is higher than that of the traditional scheduling plan and the rescheduling plan with GA. It should be mentioned that the ore tonnage of some stopes in the rescheduling plan is lower than that of the traditional scheduling plan. According to Table 14, the tonnage of ore finally hauled to the ore pass in the rescheduling plan is higher than the ore tonnage of the traditional scheduling plan. Compared to the

traditional scheduling plan, the rescheduling plan with WCA has a higher completion rate of 83.40%, which is higher than that of the rescheduling plan with GA, while the ore grade is increased by 0.74%.

**Table 14.** The data comparison of various schemes.

| Scheme | The Total Tonnage of Ore Mined/t | Plan Completion Rate/% | The Grade Fluctuation of Daily Ore Removal/% |
|---|---|---|---|
| Traditional scheduling plan | 3118.52 | 76.09 | 0 |
| Rescheduling plan with GA | 3324.68 | 81.71 | 0.57 |
| Rescheduling plan with WCA | 3418.14 | 83.40 | 0.74 |

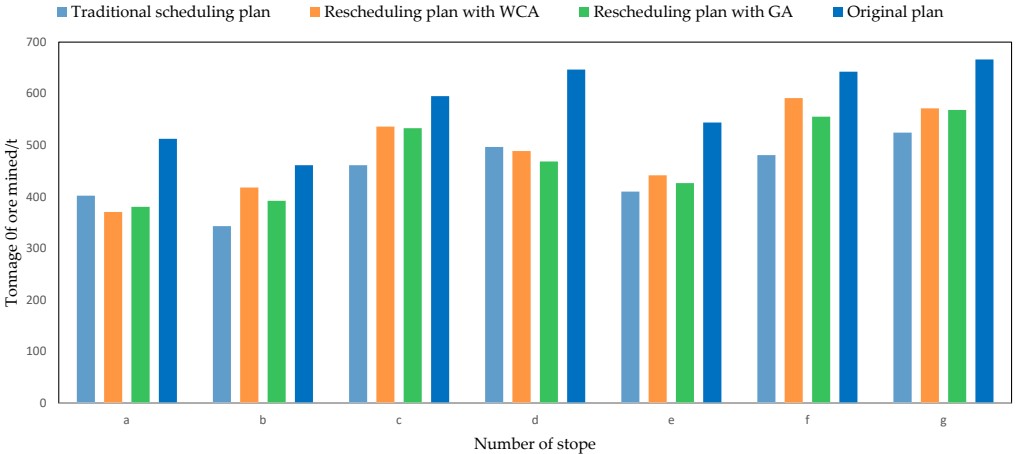

**Figure 8.** The amount of ore mined according to each scheme.

## 5. Discussion

Compared with other algorithms, WCA can find the optimal solution relatively quickly due to its ability to search at multiple points simultaneously. However, WCA is mostly used to solve the problem with continuous variables. The coding method of GA can intuitively display the scheduling decision scheme. Through performance testing, it is found that its solving ability is weak in that of WCA. In order to solve the problem of discrete coding, it is necessary to redefine the various behaviors of the individual artificial wolves.

To successfully prepare the rescheduling plan and simplify the problem, several assumptions are made in this paper. The mathematical model constructed in this study can be used in actual production. When the amount of haulage equipment is not increased, the model can be used to obtain a multi-equipment rescheduling plan, which can guide mining production. It can be considered to improve the model when spare equipment is involved. In addition, the construction method of this model can also be used for the rescheduling of other types of equipment, and the objective functions and constraints in the model may be slightly different. However, due to many factors affecting the operation of mining equipment, it is relatively difficult to predict its breakdowns. Therefore, accurately predicting the breakdown time of equipment will become one of the key aspects for the accurate preparation of a rescheduling plan, as well as one of the important research directions in the future. In addition to the breakdown time prediction, the preparation of the equipment rescheduling plan conducted in this paper is a type of static preparation method. This type belongs to the category of short-term planning and can be used as preparation for subsequent research. However, the actual equipment operation plan is dynamic, i.e., it can be adjusted in real-time. Therefore, converting a static rescheduling plan

into dynamic real-time breakdown dispatching also represents an important research direction. In addition, when carrying out the dynamic real-time breakdown dispatching of equipment, resources and environmental costs can be optimized as the targets. The aforementioned can be done to reduce environmental pollution and the loss of natural resources, as well as to promote green mine production and construction.

## 6. Conclusions

In this paper, a model of the haulage equipment rescheduling plan based on the random simulation of equipment breakdowns was established. The model was optimized by improving the wolf colony algorithm. The following conclusions are provided:

1.  According to the actual statistics of the number of haulage breakdowns in the mine, the breakdown probability can be calculated. Then, a random breakdown model, which is used to simulate the time of breakdown occurrence and maintenance end, is constructed by generating random numbers.
2.  When breakdown occurrence and maintenance end times are known, the breakdown period is divided into five inserting rescheduling periods by taking the crossing point between the breakdown time of each piece of equipment and the time when the breakdown occurs and maintenance ends. Then, during each rescheduling period, the rescheduling plan model is constructed based on random breakdown simulation. The main goal is to minimize the grade fluctuation in each period, while the secondary goal is to maximize the completion rate of the ore mining plan.
3.  The chromosome coding method of the genetic algorithm is used to improve the form of the individual population according to the wolf colony algorithm. The wandering step length, the rushing step length, the sieging step length, and the update method of the individual position of the wolf colony are refined to adapt to the solution of the rescheduling plan model. Finally, a wolf colony optimization algorithm is formed via string encoding. The algorithm has the characteristics of strong global searchability, fast convergence, and high robustness.
4.  The rescheduling plan optimized by the wolf colony algorithm is compared with the traditional scheduling planning. The results indicate that, when the equipment breaks down, the reprepared scheduling plan has fewer fluctuations in the grade of daily ore removal (which is increased by 0.74%). The completion rate is increased by 7.31%, while the quality of the ore is simultaneously improved.

**Author Contributions:** Conceptualization, N.L.; data curation, H.Y.; formal analysis, S.F.; funding acquisition, N.L. and M.J.; investigation, Q.W. and T.L.; methodology, N.L. and M.J.; resources, H.Y., S.Z., and L.W.; supervision, N.L.; validation, S.F. and T.L.; Writing—original draft, S.F.; writing—review and editing, T.L., H.Y., Q.W., M.J., and L.W. All authors have read and agreed to the published version of the manuscript.

**Funding:** This research is supported by the National Key R&D Program of China (Grant No. 2019YFC0605304).

**Institutional Review Board Statement:** Not applicable.

**Informed Consent Statement:** Not applicable.

**Data Availability Statement:** Not applicable.

**Acknowledgments:** Thanks to the relevant departments for funding and the mining companies that conducted the experiment.

**Conflicts of Interest:** The authors declare no conflict of interest.

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
