# Peer review of "Rescheduling Plan Optimization of Underground Mine Haulage Equipment Based on Random Breakdown Simulation"

_sustainability, doi:10.3390/su14063448_

Round 1

Reviewer 1 Report

The paper deals with the optimization of improving the haulage equipment rescheduling plan, based on the random simulation of equipment breakdowns using the “wolf colony” algorithm .

In the underground ore mines, as the analyzed one, the extraction is performed in many workfaces , from different veins, with different average grades. When a breakdown occurs, related to the haulage equipment, the
initial scheduling plan of haulage must be changed, in order to maintain the average ore grade of the output and the production capacity.

These two criteria are used for optimization, together with other constrains.
The rescheduling plan calculated by the optimization algorithm is compared with the traditional one, and results that the total ore tonnage of each stope in the rescheduling plan is higher than that of the traditional scheduling
plan.

The paper is interesting, with practical usefulness, the model is a little bit complicated, despite the large number of variables involved. (Not enough letters and indexes available). However, the results of the case study reveals an improvement of solving such a problem. Perhaps an example with less stopes could be more explicit for the method.

Author Response

Thank you very much for your comments on this paper. Thank you for your support of our team's work. We will make further efforts in this field.

Reviewer 2 Report

First of all, I should say that many studies have been missed in the literature review. For example, see:

https://www.sciencedirect.com/journal/computers-and-operations-research/vol/115/suppl/C

 In addition, it is not clear what the research gaps are. And the authors fail to explain the contributions of this study clearly.

The introduction of the paper is very confusing, and there is no logic in the introduction, it seems the authors merged some arbitrary sentences without any aim. Somewhere they talk about big data, somewhere about sustainability,….. it is not clear to see the logic of the introduction.

The problem statement is very bad. The model in section 2.2.4 is not a mathematical model!! Where is the mathematical models including constraints, variables,…..

While the breakdown of the machines has been considered as one of the contributions of this study, this aspect of the paper is based on a simple assumption. While there are many studies in the literature about the reliability of machines and equipments in mining and construction sites.

Bt my main concern about this paper is the solution method and the presentation of the outcome of the algorithms. First of all, I should say that there are many strong metaheuristic algorithms in the literature. Why this algorithm? Employing an obscure metaheuristic to a new scheduling problem can not be considered as a contribution!!!

In addition, when a metaheuristic algorithm is employed in a problem, some other powerful algorithms should be used to show the superiority of the proposed method. And the results should be supported by statistical tests.

The presentation of section 4 should entirely be revised. This section is very bad-written.

Author Response

Thank you very much for your comments on this paper. We read your comments carefully and then revised the paper. Please review the revised paper.

  1. In the Introduction, further revisions have been made.

  1. In section 2.2.4, there are two types of mathematical models, one is the inserting rescheduling planning model, and the other is the complete rescheduling model. The objective functions of the two are the same, but the constraints are different. Select corresponding constraints according to different types of rescheduling periods.

  1. The main research of this paper is about the preparation of rescheduling plan when considering equipment breakdown, and a method of rescheduling plan preparation is proposed, which can better arrange production and improve the overall income of the mine. In order to adapt to the solution of the problem, part of the algorithm is improved. The purpose of improving algorithm is to get the solution of the problem better, and to prove the feasibility of the rescheduling method.

  1. In the section 4, I have added some explanations and changed the order of some figures and tables.

Reviewer 3 Report

The presented article is devoted to optimizing the planning of production processes for ore mining, taking into account the failure of equipment. This problem is relevant, since the correct accounting for equipment failure affects the technical and economic indicators of the adopted mining method.

The authors used new, creative and interesting research methods for science and showed their effectiveness. The research material left a good impression. The use of the considered research methods in mines to optimize production planning allows the development of intelligent mining.

However, having carefully studied the material of the research, there were several recommendations for improving and clarifying its presentation.

  1. The authors are recommended to add a few sentences in the introduction about what was still insufficiently studied by previous scientists? What factors are not taken into account in production planning? What area of the problem requires further study.
  2. In fig. 1 and fig. 4 should explain signatures #1, #2 and #3. The timeline is also interpreted in the caption of the figure, although it is not illustrated in the figure.
  3. For the stope faces of which mine are the initial ore grade data taken? If this is the theoretical data of the authors, this should be noted. This applies to the data in tables 1, 2, 3 and 5.
  4. In the discussion section, I would like the authors to specifically formulate the scientific significance and value of the results obtained (new model, new approaches and other). This will emphasize the novelty and originality of the study.
  5. The list of references contains citations of only Chinese scientists. Although scientists from other countries are also dealing with a similar problem. For example, methods of neural network planning. It is recommended to expand the geography of citations.

Author Response

Thank you very much for your comments on this paper. We read your comments carefully and then revised the paper. Please review the revised paper.

  1. In the Introduction, further revisions have been made.

  1. In Figures 1 and 4, the timeline is further illustrated in the figure, as is the equipment number.

  1. The data in Tables 1, 2, 3, and 5 are theoretical data and have been marked in the table.

  1. The Discussion section has been supplemented.

  1. I have added some citations by scientists from other countries in the References.

Round 2

Reviewer 2 Report

unfortunately, the authors provide an unprofessional response sheet and they didn't address my comments. they evade addressing some of my comments particular those that are related to the solution method. I can not accept this paper at all and I have to reject it.

Author Response

Dear Reviewer,

Thank you very much for your valuable comments on this paper. These comments are particularly helpful in improving the quality of this paper. Due to the incomplete work in the early stage, there are some problems in this paper. I would like to make an in-depth study in the next paper.

In the latest minor revision, I have revised the following contents:

(1) In Section 2.2, I have revised the rescheduling plan model, parameters and variables. The rescheduling plan model are divided into different rescheduling plan model, and the variables in each objective are been revised in Section 2.2.4.

(2) In Section 3, Section 3.1 is added aiming to compare the performance of the wolf colony algorithm with other algorithms.

(3) In Section 4, since the encoding adopts the string encoding method, the presentation of the plan scheme is in the form of string, which is difficult for readers who do not know this method to understand, so a new reinterpretation about the content in Table 4 is given.

(4) In Section 4.4, the comparison between the optimization results of WCA and GA is added.

(5) In Discussion, the new discussion about WCA is added.

We deeply appreciate your consideration of our manuscript, and we look forward to receiving favorable comments from you.

Thank you for your consideration of this manuscript.

Sincerely,

Ning Li, Shuzhao Feng, Tao Lei, Haiwang Ye, Qizhou Wang, Liguan Wang, Mingtao Jia

E-mail: 13875910191@163.com
